# Precoding Design for Energy Efficiency Maximization in MIMO Half-Duplex Wireless Sensor Networks with SWIPT

**DOI:** 10.3390/s19224923

**Published:** 2019-11-12

**Authors:** Liang Xue, Jin-Long Wang, Jie Li, Yan-Long Wang, Xin-Ping Guan

**Affiliations:** 1School of Information and Electrical Engineering, Hebei University of Engineering, Handan 056038, China; xueliang19491982@163.com; 2School of Mechanical and Electrical Engineering, Handan University, Handan 056038, China; lijielijie1982@163.com; 3Key Laboratory of Trustworthy Distributed Computing and Service, Ministry of Education, School of Information and Communication Engineering, Beijing University of Posts and Telecommunications (BUPT), Beijing 100876, China; wylyanlong@gmail.com; 4Key Laboratory of System Control and Information Processing, Ministry of Education, Shanghai Jiaotong University, Shanghai 200240, China; xpguan@sjtu.edu.cn

**Keywords:** energy efficiency, multiple-input multiple-output, precoding matrices, simultaneous wireless information and power transfer, wireless sensor networks

## Abstract

This paper explores the energy efficiency (EE) maximization problem in single-hop multiple-input multiple-output (MIMO) half-duplex wireless sensor networks (WSNs) with simultaneous wireless information and power transfer (SWIPT). Such an energy efficiency maximization problem is considered in two different scenarios, in which the number of energy-harvesting (EH) sensor nodes are different. In the scenario where the single energy-harvesting sensor node is applied, the modeled network consists of two multiple-antenna transceivers, of which the energy-constrained energy-harvesting sensor node harvests energy from the signals transmitted from the source by a power splitting (PS) scheme. In the scenario of multiple EH sensor nodes, *K* energy-constrained sensor nodes are applied and the same quantity of antennas are equiped on each of them. The optimization problem is formulated to maximize the energy efficiency by jointly designing the transceivers’ precoding matrices and the PS factor of the energy-harvesting sensor node. The considered constraints are the required harvested energy, the transmission power limit and the requirement on the data rate. The joint design of the precoding matrices and the PS factor can be formulated as an optimization problem, which can be transformed into two sub-problems. An alternating algorithm based on Dinkelbach is proposed to solve the two sub-problems. The convergence of the proposed alternating algorithm, the solution optimality and the computational complexity are analyzed in the paper. Simulation results demonstrate the convergence and effectiveness of our proposed algorithm for realizing the maximum energy efficiency.

## 1. Introduction

The emergence of tens of billions of intelligent Internet-of-Things (IoT) devices have attracted serious concerns about their-energy waste and the radiation pollution. Green communications technology opens a new road for energy exploitations and gradually promotes its popularity nowadays [1]. Simultaneous wireless information and power transfer (SWIPT) was first proposed in [2], and can be regarded as the candidate technology that meets the idea of green communications because it aims at realizing a high data rate and at enabling energy reuse. It enables the energy-constrained nodes to harvest recyclable energy and to decode information on the same radio frequency (RF) signals. The tendency inspires a feasible scheme to extend the lifetime of the energy constrained nodes in wireless sensor networks. In SWIPT, the EH receiver and information-decoding (ID) receiver can be functioned in two modes, i.e., co-located and separated [3]. For the co-located receiver, Zhang, R. et al. in [4] proposed two efficient information transmission mechanisms, known as power-splitting (PS) and time-switching (TS). In the PS mechanism, the co-located receiver divides the received signal power into two power streams in an appropriate proportion, one for information decoding and the other for energy harvesting. In the TS mechanism, the co-located receiver is designed to switch its modes from energy harvesting to information decoding in different time slots.

Following these pioneering works, SWIPT began its applications in the single-input single-output (SISO) system [5] and is now developed to be used in scenarios, e.g., the multiple-input single-output (MISO) systems [6,7,8], multiple-input multiple-output (MIMO) systems [9,10,11], broadcast systems [12], orthogonal frequency-division multiplexing (OFDM) relay systems [13], etc. In [8], the two-way multi-antenna relay networks are considered, where the beamforming of relay antennas is designed to maximize the sum rate under the energy constraint. Jiang, R. et al. in [12] studied the broadcast networks system, where the transmitting power of a hybrid access point (H-AP) equipped with multiple antennas is minimized under constraints such as the minimum signal-to-interference-plus-noise ratio (SINR) and the power harvested at each user with a single antenna. Shen, Y. et al. in [13] investigated the resource optimization problem in TS relay networks, which take into account a realistic energy consumption model. It is worth noting that MIMO technology has been widely used in modern mobile communication networks due to its considerable diversity gain and effective resistance to path loss. The combination of MIMO and SWIPT is the cutting edge research direction at present. Following this research trend, Zhang, R. et al. in [4] studied the rate-energy trade-off region under the constraint of nonlinear energy harvesting. By giving an actual EH model, the optimal trade-off between the harvested energy and data rate was explored in [4]. By optimizing the precoding at the source and the relay, the achievable transmission rate with TS or with PS schemes, respectively, in a two-hop decode-and-forward (AF) relay network are investigated in [9]. In [11], the two-user MIMO interference channel is proposed and secure communication is realized by designing optimal precoding on transmitters.

However, few of the above studies were concerned with the energy efficiency, but energy efficiency has developed to be the performance metric of energy-constrained networks due to the requirement on energy management [14,15,16,17]. Energy efficiency is defined as the ratio of the network sum rate to the total energy consumption. In [10], all of the desired receivers decode information and harvest energy in a point-to-point communication system with a PS scheme. The desired receivers can achieve the optimal energy efficiency by jointly optimizing the transmission precoding matrix, the artificial noise covariance matrix and the power splitting ratios. Not only that, the probability of eavesdropping is also reduced. In [14], Ng, D.W.K. et al. study the resource allocation algorithm design for OFDM downlink systems to maximize the energy efficiency. An alternating iteration algorithm is developed in [15] to infer the optimal precoding matrix at the relay and transmitters in the two-way relay networks. Zhou, X. et al. in [16] studied a two-way amplify-and-forward relay networks. An iterative algorithm based on the constrained concave convex procedure (CCCP) is proposed to jointly design the sources’ and relay’s precoding matrices and the PS ratios. The iterative algorithm maximizes the energy efficiency of the network. Similarly, the energy efficiency is maximized in [17], in which a distributed antenna system is studied. Through adjusting the PS ratios, this enables the IoT devices to coordinate the energy harvesting and information decoding. The users of the wireless powered communication networks in [18] can harvest energy from dedicated power station, which ensures that the information can be transmitted to the information reception station. Then the authors Wu, Q. et al. in [18] maximize the energy efficiency of the system via the joint optimization of slot allocation and power control. A utility function is proposed in a MISO system [19] with the purpose of maximizing the ratio of the achieved utility to the total power consumption. A MISO downlink data transmission link is considered in [20], in which the zero-forcing (ZF) algorithm was applied by the base station. The energy efficiency of the system is maximized through joint optimization of beamforming and power splitting.

The realizations of the above works all benefited from the development of EH technologies. They paved the way for filling up the deficiencies of the energy constrained networks. In recent years, EH technologies continue to be developed, i.e., recyclable energy can be harvested from either the surrounding environment or the dedicated RF signals and the lifetime of wireless networks can be improved [21]. Gui, L. et al. in [22] propose a new secure system model with both fixed and mobile jammers to guarantee secrecy in the worst-case scenario, where all jammers are thought to have EH capability. In [23], sensor nodes harvest energy from mobile phones’ RF signals and, in order to acquire available power use from the ambient mobile phones, a contract theory based incentive scheme is investigated. With the linear EH model, the harvested energy in unit time is usually proportional to the power of the received RF signals. By analyzing the experimental results, Boshkovska, E. et al. in [24] validate that for the EH device there exists a nonlinear relationship between the power of output direct current (DC) and the power of input RF signals. However, as in the conclusion given in [17], the linear EH model is still applicable for the low input power devices. The main reasons are described as follows. Firstly, the nonlinear EH model can be approximated to be a piecewise linear model when the EH devices are assigned with relatively low or high input power, separately. With such assumption, the linear EH model has acceptable accuracy. Secondly, the intelligent IoT devices are usually exposed in an electromagnetic surrounding filled with low power RF signals.

In this paper, the maximization of energy efficiency is studied in MIMO half-duplex SWIPT wireless sensor networks with a linear EH model. In fact, some efforts have been made to achieve highly efficient utilization of network resources in WSNs. Guo, W. et al. in [25] propose a soft real-time fault-tolerant task allocation algorithm (FTAOA) for WSNs to support the fault tolerance mechanism to improve resource utilization. To reduce the total traffic cost, Yao, Y. et al. in [26] develop the data collection protocol EDAL. In [27], a node scheduling strategy is proposed to reduce the number of active nodes and the amount of messages in the network. The strategy proposed in [27] reduces the energy consumption and improves the network lifetime. Moreover, in order to ensure the communication reliability in wireless networks, the proposal in [28] studies the distributed scheduling problem, by which the scheduling cycle length is to be minimized.

This paper incorporates SWIPT with WSNs, in which the energy efficiency maximization problem is studied. The main contributions of this paper are summarized as follows:The EE maximization problem in the SWIPT-based MIMO wireless sensor network is considered by employing the PS mechanism. Our goal is to maximize the energy efficiency by jointly optimizing the power splitting factor and the precoding matrices of transceivers. Moreover, the upper limit on the power supply and the required data rate for quality of service (QoS) guarantee is bounded as the constraints.For the proposed optimization problem, the objective function is non-convex. The eigenvalue decomposition and singular value decomposition are applied to reduce the dimensionality. Meanwhile, the original coupling problem can be transformed to two sub-problems. For these two sub-problems, an alternating algorithm based on Dinkelbach is used to guarantee the convergence.Excluding the convergence analysis, the computational complexity is given in the paper and the feasibility of our proposal is also verified by simulation results. The performance comparisons between our proposal and the other selected baseline algorithms validate the effectiveness of our proposal.

The rest of the paper is organized as follows: Section 2 introduces the system model. In Section 3, the optimization problem of power allocation for EE maximization is formulated and then an alternating algorithm based on Dinkelbach is introduced to solve the problem. The convergence and computational complexity of the algorithm are also analyzed in Section 3. Section 4 shows the extended energy efficiency maximization problem in the scenario of the multiple EH sensor nodes used. Section 5 shows the simulation results and performance comparisons. Section 6 concludes the paper.

## 2. Single EH Sensor Nodes System Model

### 2.1. Notations

This paper specifies that the bold italics uppercase and the bold italics lowercase letters denote matrices and vectors, respectively. The operators, i.e., $A^{H}$, $A^{-1}$, $A^{+}$, Tr(A), $\left|A\right|$ and $∥A∥$ represent conjugate transpose, inverse, pseudo-inverse, trace, determinant and Frobenius norm of the matrix A, respectively. A⪰0 means that A is positive semidefinite. $C^{N\times M}$ is defined as the space of N×M matrices with complex entries. $I_{N}$ denotes the N×N identity matrix and 0 denotes zero matrix.

### 2.2. System Model

The MIMO half-duplex wireless sensor network is considered in Figure 1, which contains two transceivers, i.e., the source and the EH sensor node, equipped with $N_{1}$ and $N_{2}$ antennas, respectively. The source node has a persistent energy supplement, whereas the the EH sensor node is an energy-constrained node. To maintain the EH sensor node’s survival, the PS scheme is adopted at the sensor node to harvest energy from the source’s RF signals, which are divided into two power flows by the power splitting scheme. α(0≤α≤1) is defined as the power splitting factor, by which the α portion of the received signals is used for the ID receiver and the (1−α) portion of the received signals is used for the RF energy conversion circuit. Following the same channel models given in [12,17], the frequency flat slow block fading channel model that obeys Rayleigh distribution is adopted in this paper. The channel state information (CSI) of all the links are also known beforehand according to the historical statistics.

#### 2.2.1. Source—EH Sensor Node Communication

The two-way data transmission between the source and the sensor node occurs on two consecutive equal time slots, just as shown in Figure 1. During the time slot 1, the source transmits its signals to the sensor node and the received signals at the sensor node are given as
(1)$$\begin{array}{c} y_{2}=\mathrm{HF}s_{1}+n_{2}, \end{array}$$
where $s_{1}\in C^{N_{1}\times 1}$$\left(s_{1}∼CN\left(0,1\right)\right)$ is the symbol vector generated at the source, $F\in C^{N_{1}\times N_{1}}$ denotes the precoding matrix of the source, $H\in C^{N_{2}\times N_{1}}$ denotes the channel coefficient matrix from the source to the sensor node. The channel can be assumed to be quasi-static [29], so that the channel coefficient matrix keeps unvaried in the two time slots. $n_{2}\in C^{N_{2}\times 1}$ is the zero-mean additive white Gaussian noises (AWGN) at the sensor node with the covariance matrix $\sigma_{2}^{2}I_{N_{2}}$.

By the power splitting factor α, the received signals at the sensor node can be split into two flows. In the paper, each antenna of the sensor node is assigned with the same power splitting factor for problem simplification. Therefore, the signals for ID at the sensor node can be given as
(2)$$\begin{array}{c} y_{2}^{ID}=\sqrt{\alpha }\left(\mathrm{HF}s_{1}+n_{2}\right)+n_{2}^{ID}, \end{array}$$
where $n_{2}^{ID}∼CN\left(0,\sigma_{ID}^{2}I_{N_{2}}\right)$ is the AWGN generated by the baseband signal processing of the ID receiver, which converts the RF signals into baseband signals. Thus the achievable rate in bits/s/Hz at the sensor node can be given by
(3)$$\begin{array}{c} R_{2}=\log_{2}\left|I_{N_{2}}+\frac{\alpha \mathrm{HF}F^{H}H^{H}}{\left(\alpha \sigma_{2}^{2}+\sigma_{ID}^{2}\right)I_{N_{2}}}\right|. \end{array}$$

The data rate $R_{2}$ in Equation (Equation 3) can be achieved due to a perfect CSI being known, which can be obtained by training at either the uplink or the downlink using pilot signals or by using the 1-bit feedback information [17].

On the one hand, the signals for EH are shown as in Equation (Equation 4)
(4)$$\begin{array}{c} y_{2}^{EH}=\sqrt{1-\alpha }\left(\mathrm{HF}s_{1}+n_{2}\right). \end{array}$$

Based on Equation (Equation 4), the harvested energy at the sensor node can be deduced as in Equation (Equation 5) [17]
(5)$$\begin{array}{c} E_{EH}=\eta \left(1-\alpha \right)Tr\left(\mathrm{HF}F^{H}H^{H}+\sigma_{2}^{2}I_{N_{2}}\right), \end{array}$$
where $\eta \in \left[0,1\right]$ is the EH efficiency that reflects the proportion of the harvested energy from the received RF signals. The considered MIMO sensor network always operates at low input power at the sensor node, the linear EH model can be still applicable to the MIMO sensor network.

During time slot 2, the sensor node is powered by the harvested energy and transmits signals to the source. The received signal at the source can be expressed as
(6)$$\begin{array}{c} y_{1}=H^{H}Ds_{2}+n_{1}, \end{array}$$
where $s_{2}\in C^{N_{2}\times 1}$ denotes the unit-norm information symbol vector generated by the sensor node, $D\in C^{N_{2}\times N_{2}}$ denotes the precoding matrix at the source and $n_{1}∼CN\left(0,\sigma_{1}^{2}I_{N_{1}}\right)$ represents the AWGN with $n_{1}\in C^{N_{1}\times 1}$. Similar to the achievable rate shown as Equation (Equation 3), the achievable rate in bits/s/Hz at the source can be expressed as Equation (Equation 7).
(7)$$\begin{array}{c} R_{1}=\log_{2}\left|I_{N_{1}}+\frac{1}{\sigma_{1}^{2}}H^{H}DD^{H}H\right|. \end{array}$$

The rate $R_{1}$ can be obtained just following the data rate given in Equation (Equation 3). Assume that the the time slot for transmission at the source or the sensor node occupies a 1/2 time frame; thus, the total amount of the transmitted data equals $\frac{T}{2}\left(R_{2}+R_{1}\right)$ in bits/Hz.

#### 2.2.2. Energy Efficiency

The energy efficiency (bits/Hz/J) in the paper is defined as the ratio of total number of bits conveyed in networks to the networks’ energy consumption. According to Equations (Equation 1) and (Equation 6), the transmission power of the source and the sensor node denoted as $P_{1}$ and $P_{2}$ can be obtained as follows
(8)$$\begin{array}{c} P_{1}={∥F∥}^{2}=Tr\left(Q_{1}\right), \end{array}$$
(9)$$\begin{array}{c} P_{2}={∥D∥}^{2}=Tr\left(Q_{2}\right), \end{array}$$
where $Q_{1}=FF^{H}$, $Q_{2}=DD^{H}$. Therefore, the networks’ energy consumption within the considered time slots can be given by Equation (Equation 10) [16].
(10)E(α,Q1,Q2)=T2(P1+P2)(P1+P2)ςς+Pc,
where ς∈(0,1] is the drain efficiency for the power amplifiers of transceivers. Assume that the source and the sensor node have the same drain efficiency, and are set as 100%, i.e., ς=1. The factor $\frac{T}{2}$ is due to the power consumption, i.e., $P_{1}$ and $P_{2}$ separately occurred in two time slots. The period of each time slot is 1/2 unit time frame. $P_{c}$ denotes the sum of circuit power consumption at the source and the sensor node and $P_{c}=P_{c1}+P_{c2}$, where $P_{c1}$ and $P_{c2}$ represent the circuit power consumption of the source and the sensor node, respectively, The circuit power is used to enable transceivers to transmit and receive the signals.

The harvested energy during the time slot should maintain the sensor node’s survival and hence is required to meet the minimum requirement for the power needs, therefore the constraint in Equation (Equation 11) can be established as
(11)$$\begin{array}{c} E_{EH}\geq P_{2}+P_{c2}. \end{array}$$

According to the definitions given above, the EE in the paper can be formulated as follows
(12)$$\begin{array}{c} EE=\frac{\frac{T}{2}\left(R_{2}+R_{1}\right)}{E(\alpha ,Q_{1},Q_{2})}. \end{array}$$

## 3. Problem Formulation and Solution

### 3.1. Optimization Problem Formulation

This paper aims at maximizing the network energy efficiency. It considers the joint optimization of the power splitting factor α and the precoding matrices $Q_{1}$ and $Q_{2}$. The constraints such as the upper limit on the transmission power, the required minimum data rate for the QoS guarantee and the semi-definite attribute of the precoding matrices are taken into account in the problem formulation. To enable the passive sensor node to receive sustaining energy for its operation, the harvested energy during the two time slots should cover the sensor node’s lowest power needs. Therefore, the problem for maximizing energy efficiency in the considered network can be formulated as **P1**,
(13a)$$\begin{array}{cc} P1: & \max_{\alpha ,Q_{1},Q_{2}}\overset{}{\underset{}{}}\frac{\frac{T}{2}\left(R_{2}+R_{1}\right)}{E(\alpha ,Q_{1},Q_{2})} \end{array}$$
(13b)$$\begin{array}{cc} \; & s.t.\;E_{EH}\geq P_{2}+P_{c2} \end{array}$$
(13c)$$\begin{array}{cc} \; & Tr\left(Q_{1}\right)\leq {\bar{P}}_{\max}^{1} \end{array}$$
(13d)$$\begin{array}{cc} \; & Tr\left(Q_{2}\right)\leq {\bar{P}}_{\max}^{2} \end{array}$$
(13e)$$\begin{array}{cc} \; & R_{2}\geq {\bar{r}}_{\min}^{2} \end{array}$$
(13f)$$\begin{array}{cc} \; & R_{1}\geq {\bar{r}}_{\min}^{1} \end{array}$$
(13g)$$\begin{array}{cc} \; & 0\leq \alpha \leq 1,Q_{1}⪰0,Q_{2}⪰0, \end{array}$$
where ${\bar{P}}_{\max}^{1}$ and ${\bar{P}}_{\max}^{2}$ stand for the upper limits on the transmission power of the source and the sensor node, respectively, ${\bar{r}}_{\min}^{1}$ and ${\bar{r}}_{\min}^{2}$ represent data rate threshold for the source and the sensor node, respectively, $Q_{1}⪰0$ and $Q_{2}⪰0$ imply that the precoding matrices $Q_{1}$ and $Q_{2}$ are positive semi-definite. The constraint of Equation (13b) guarantees that the energy harvested at the sensor node can cover its lowest power needs.

### 3.2. Proposed Optimal Solution

The formulated problem P1 is non-convex due to the parameters α and precoding matrix $Q_{1}$ being highly coupled in both the objective function and the constraints of Equations (13b) and (13e). Therefore, it is generally quite difficult to solve the non-convex optimization problem. As far as we know, there exist no practical methods that can guarantee to converge to the global optimal solution. In this section, a diagonalization method with low complexity is used to solve the problem P1.

The precoding matrix $Q_{1}$ can be decomposed as $Q_{1}=U_{Q_{1}}\Lambda_{Q_{1}}U_{Q_{1}}^{H}$ through eigenvalue decomposition (EVD). Similarly, the precoding matrix $Q_{2}$ can also be decomposed as $Q_{2}=U_{Q_{2}}\Lambda_{Q_{2}}U_{Q_{2}}^{H}$ by EVD. If singular value decomposition (SVD) is applied to the channel coefficient matrix H, then H can be decomposed as $H=U_{H}\Lambda_{H}^{\frac{1}{2}}V_{H}^{H}$. Based on these decomposition expressions, the problem P1 can be transformed to problem P2 shown as at the top of this page,
$$\begin{array}{cc} \; & P2:\max_{\alpha ,\Lambda_{Q_{1}},\Lambda_{Q_{2}}}\frac{\log_{2}\left|I_{N_{2}}+\frac{\alpha }{\alpha \sigma_{2}^{2}+\sigma_{ID}^{2}}U_{H}\Lambda_{H}^{\frac{1}{2}}V_{H}^{H}U_{Q_{1}}\Lambda_{Q_{1}}U_{Q_{1}}^{H}V_{H}\Lambda_{H}^{\frac{1}{2}H}U_{H}^{H}\right|}{Tr\left(U_{Q_{1}}\Lambda_{Q_{1}}U_{Q_{1}}^{H}\right)+Tr\left(U_{Q_{2}}\Lambda_{Q_{2}}U_{Q_{2}}^{H}\right)+P_{c}} \end{array}$$
(14a)$$\begin{array}{cc} \; & \;\;\;\;\;\;\;\;\;\;\;\;\;\;\;\;\;\;+\frac{\log_{2}\left|I_{N_{1}}+\frac{1}{\sigma_{1}^{2}}V_{H}\Lambda_{H}^{\frac{1}{2}H}U_{H}^{H}U_{Q_{2}}\Lambda_{Q_{2}}U_{Q_{2}}^{H}U_{H}\Lambda_{H}^{\frac{1}{2}}V_{H}^{H}\right|}{Tr\left(U_{Q_{1}}\Lambda_{Q_{1}}U_{Q_{1}}^{H}\right)+Tr\left(U_{Q_{2}}\Lambda_{Q_{2}}U_{Q_{2}}^{H}\right)+P_{c}} \end{array}$$
(14b)$$\begin{array}{cc} \; & s.t.\;\eta \left(1-\alpha \right)Tr\left(U_{H}\Lambda_{H}^{\frac{1}{2}}V_{H}^{H}U_{Q_{1}}\Lambda_{Q_{1}}U_{Q_{1}}^{H}V_{H}\Lambda_{H}^{\frac{1}{2}H}U_{H}^{H}+\sigma_{2}^{2}I_{N_{2}}\right)\geq Tr\left(U_{Q_{2}}\Lambda_{Q_{2}}U_{Q_{2}}^{H}\right)+P_{c2} \end{array}$$
(14c)$$\begin{array}{cc} \; & Tr\left(U_{Q_{1}}\Lambda_{Q_{1}}U_{Q_{1}}^{H}\right)\leq {\bar{P}}_{\max}^{1} \end{array}$$
(14d)$$\begin{array}{cc} \; & Tr\left(U_{Q_{2}}\Lambda_{Q_{2}}U_{Q_{2}}^{H}\right)\leq {\bar{P}}_{\max}^{2} \end{array}$$
(14e)$$\begin{array}{cc} \; & \log_{2}\left|I_{N_{2}}+\frac{\alpha }{\alpha \sigma_{2}^{2}+\sigma_{ID}^{2}}U_{H}\Lambda_{H}^{\frac{1}{2}}V_{H}^{H}U_{Q_{1}}\Lambda_{Q_{1}}U_{Q_{1}}^{H}V_{H}\Lambda_{H}^{\frac{1}{2}H}U_{H}^{H}\right|\geq {\bar{r}}_{\min}^{2} \end{array}$$
(14f)$$\begin{array}{cc} \; & \log_{2}\left|I_{N_{1}}+\frac{1}{\sigma_{1}^{2}}V_{H}\Lambda_{H}^{\frac{1}{2}H}U_{H}^{H}U_{Q_{2}}\Lambda_{Q_{2}}U_{Q_{2}}^{H}U_{H}\Lambda_{H}^{\frac{1}{2}}V_{H}^{H}\right|\geq {\bar{r}}_{\min}^{1} \end{array}$$
(14g)$$\begin{array}{cc} \; & 0\leq \alpha \leq 1, \end{array}$$
where $\Lambda_{Q_{i}}=diag\left(\lambda_{1,Q_{i}},\lambda_{2,Q_{i}},\lambda_{3,Q_{i}},\ldots \lambda_{N_{i},Q_{i}}\right)$ with $\lambda_{1,Q_{i}}\geq \lambda_{2,Q_{i}}\geq \lambda_{3,Q_{i}}\geq \ldots \lambda_{N_{i},Q_{i}}\geq 0$(i=1,2) and $\Lambda_{H}^{\frac{1}{2}}=diag\left(\sqrt{\lambda_{1,H}},\sqrt{\lambda_{2,H}},\sqrt{\lambda_{3,H}},\ldots \sqrt{\lambda_{\min\{N_{1},N_{2}\},H}}\right)$ with $\lambda_{1,H}\geq \lambda_{2,H}\geq \lambda_{3,H}\geq \ldots \lambda_{\min\{N_{1},N_{2}\},H}\geq 0$. $U_{H}\in C^{N_{2}\times \min\left\{N_{1},N_{2}\right\}}$ and $V_{H}\in C^{N_{1}\times \min\left\{N_{1},N_{2}\right\}}$ are all the unitary matrices, which are constituted by the orthogonal unit norm column vectors.

As to the maximization of network energy efficiency, the best precoding matrices $Q_{1}^{*}$ and $Q_{2}^{*}$ make the denominator of the objective function Equation (14a) a minimum and if they simultaneously maximize the numerator, then the matrices $Q_{1}^{*}$ and $Q_{2}^{*}$ can also achieve the maximum of Equation (14a). Inspired by the form of this fraction, if precoding matrices $Q_{1}$ and $Q_{2}$ are individually quantified with a definite value, the denominator of EE can correspondingly obtain a definite value. The numerator is the sum rates of two transceivers and no coupling relations among $Q_{1}$ and $Q_{2}$ appear in both expressions of $R_{1}$ and $R_{2}$. When $Q_{1}$ and $Q_{2}$ are definite, the maximum of the objective function of P2 can be obtained only if the function value of the numerator is maximized. The sum rates, i.e., $R_{1}$ plus $R_{2}$ given in the numerator, can be maximized by separating the sum rates into two individual items, i.e., $R_{1}$ and $R_{2}$, and then by maximizing each of them. As the fundamentals for solving P2, two basic operation laws should be introduced at first.

**Operation law 1:** For any two matrices $A\in C^{m\times n}$ and $B\in C^{n\times m}$, and $\det\left(I+\mathrm{AB}\right)=\det\left(I+\mathrm{BA}\right)$ is always satisfied.

**Operation law 2:** For any two matrices $A\in C^{m\times n}$ and $B\in C^{n\times m}$, and Tr(AB)=Tr(BA) is always true. 

According to the operation laws 1 and 2, the objective function of P2 can be restructured as in Equation (15),
(15)$$\begin{array}{c} EE=\frac{\log_{2}\left|I_{N_{2}}+\frac{\alpha }{\alpha \sigma_{2}^{2}+\sigma_{ID}^{2}}V_{H}\Lambda_{H}V_{H}^{H}U_{Q_{1}}\Lambda_{Q_{1}}U_{Q_{1}}^{H}\right|+\log_{2}\left|I_{N_{1}}+\frac{1}{\sigma_{1}^{2}}U_{H}\Lambda_{H}U_{H}^{H}U_{Q_{2}}\Lambda_{Q_{2}}U_{Q_{2}}^{H}\right|}{Tr\left(\Lambda_{Q_{1}}\right)+Tr\left(\Lambda_{Q_{2}}\right)+P_{c}}. \end{array}$$

Denote $\Pi_{1}=\Theta_{1}\Lambda_{Q_{1}}\Theta_{1}^{H}$ with $\Theta_{1}=V_{H}^{H}U_{Q_{1}}$ and $\Pi_{2}=\Theta_{2}\Lambda_{Q_{2}}\Theta_{2}^{H}$ with $\Theta_{2}=U_{H}^{H}U_{Q_{2}}$. According to the operation law 1, the energy efficiency shown as in Equation (15) can be further transformed to Equation (16), which can be regarded as the objective function of a newly generated optimization problem P3.
(16)$$\begin{array}{c} EE=\frac{\log_{2}\left|I_{N_{2}}\;\;+\;\frac{\alpha }{\alpha \sigma_{2}^{2}+\sigma_{ID}^{2}}\Lambda_{H}\Pi_{1}\right|\;\;+\;\log_{2}\left|I_{N_{1}}\;\;+\;\frac{1}{\sigma_{1}^{2}}\Lambda_{H}\Pi_{2}\right|}{Tr\left(\Pi_{1}\right)+Tr\left(\Pi_{2}\right)+P_{c}}. \end{array}$$
(17a)$$\begin{array}{cc} P3: & \underset{\Pi_{1},\Pi_{2},\alpha }{max}\;\frac{\log_{2}\left|I_{N_{2}}+\frac{\alpha }{\alpha \sigma_{2}^{2}+\sigma_{ID}^{2}}\Lambda_{H}\Pi_{1}\right|+\log_{2}\left|I_{N_{1}}+\frac{1}{\sigma_{1}^{2}}\Lambda_{H}\Pi_{2}\right|}{Tr\left(\Pi_{1}\right)+Tr\left(\Pi_{2}\right)+P_{c}} \end{array}$$
(17b)$$\begin{array}{cc} \; & s.t.\;\eta \left(1-\alpha \right)Tr\left(\Lambda_{H}^{}\Pi_{1}+\sigma_{2}^{2}I_{N_{2}}\right)\geq Tr\left(\Pi_{2}\right)+P_{c2} \end{array}$$
(17c)$$\begin{array}{cc} \; & Tr\left(\Pi_{1}\right)\leq {\bar{P}}_{\max}^{1} \end{array}$$
(17d)$$\begin{array}{cc} \; & Tr\left(\Pi_{2}\right)\leq {\bar{P}}_{\max}^{2} \end{array}$$
(17e)$$\begin{array}{cc} \; & \log_{2}\left|I_{N_{2}}+\frac{\alpha }{\alpha \sigma_{2}^{2}+\sigma_{ID}^{2}}\Lambda_{H}^{}\Pi_{1}\right|\geq {\bar{r}}_{\min}^{2} \end{array}$$
(17f)$$\begin{array}{cc} \; & \log_{2}\left|I_{N_{1}}+\frac{1}{\sigma_{1}^{2}}\Lambda_{H}^{}\Pi_{2}\right|\geq {\bar{r}}_{\min}^{1} \end{array}$$
(17g)$$\begin{array}{cc} \; & 0\leq \alpha \leq 1. \end{array}$$

According to the result of the Hadamard inequality [30], given that $\Pi_{1}=\Theta_{1}\Lambda_{Q_{1}}\Theta_{1}^{H}$ and $\Pi_{2}=\Theta_{2}\Lambda_{Q_{2}}\Theta_{2}^{H}$, the optimization variables $\Pi_{1}$ and $\Pi_{2}$ must be diagonal matrices. For better conveniences, their optimums in P3 can be given as $\Pi_{1}^{*}$ and $\Pi_{2}^{*}$. Furthermore, $\Pi_{1}$ and $\Pi_{2}$ are similar to the diagonal matrices $\Lambda_{Q_{1}}$ and $\Lambda_{Q_{2}}$, respectively. Therefore, $\Pi_{1}^{*}$ and $Q_{1}^{*}$ should obtain the same eigenvalues and such a conclusion applies equally well to $\Pi_{2}^{*}$ and $Q_{2}^{*}$.

**Theorem** **1.**
*For any two commutative matrices A and B, if A is positive definite Hermitian matrices and B is n×n positive semidefinite Hermitian matrices that have eigenvalues denoted as ${\left\{\omega_{i}\right\}}_{i=1}^{n}$ and ${\left\{\tau_{i}\right\}}_{i=1}^{n}$, then the function value of $\left|A+B\right|$ can be bounded by sorting their eigenvalues in multiplication calculations. For better clarity, Theorem 1 can be illustrated by a set of inequalities as in Equation (18),*
(18)$$\begin{array}{c} \prod_{i=1}^{n}(1+\frac{{\bar{\tau }}_{i}}{{\bar{\omega }}_{i}})\leq \left|I_{n}+A^{-1}B\right|\leq \prod_{i=1}^{n}\left(1+\frac{{\bar{\tau }}_{i}}{{}_{i}}\right), \end{array}$$
*where the overline and underline represent the ascending and descending order of the eigenvalues, respectively.*


**Proof.** Please refer to [15] for the proof of Property 2. □

**Corollary** **1.**
*For any two n×n matrices A and B, if A is positive semidefinite diagonal Hermitian matrices and B is positive semidefinite Hermitian matrices, the following inequality is always true*
(19)$$\begin{array}{c} \prod_{i=1}^{n}(1+{{\underline{\omega }}_{i}}^{+}{\bar{\tau }}_{i})\leq \left|I_{n}+A^{+}B\right|\leq \prod_{i=1}^{n}\left(1+{{\bar{\omega }}_{i}}^{+}{\bar{\tau }}_{i}\right), \end{array}$$
*where, ${\left\{{\omega_{i}}^{+}\right\}}_{i=1}^{n}$ represents the eigenvalues of $A^{+}$.*


Based on the result of Theorem 1 and the corollary, the EE in P3 can be maximized such that $\Pi_{1}=\Lambda_{Q_{1}}$, $\Pi_{2}=\Lambda_{Q_{2}}$. In order to achieve above equation relations, $U_{Q_{1}}=V_{H}$ and $U_{Q_{2}}=U_{H}$ must be required. Meanwhile, regarding $\Lambda_{H}$ as $A^{+}$; therefore, the problem P3 can be simplified to the problem P4 as follows
(20a)$$\begin{array}{cc} P4: & \max_{\alpha ,\lambda_{Q_{1}},\lambda_{Q_{2}}}\overset{}{\underset{}{}}\frac{\sum_{i=1}^{N_{1}}\log_{2}\left(1+\frac{\alpha \lambda_{i,H}\lambda_{i,Q_{1}}}{\alpha \sigma_{2}^{2}+\sigma_{ID}^{2}}\right)+\sum_{i=1}^{N_{2}}\log_{2}\left(1+\frac{\lambda_{i,H}\lambda_{i,Q_{2}}}{\sigma_{1}^{2}}\right)}{\sum_{i=1}^{N_{1}}\lambda_{i,Q_{1}}+\sum_{i=1}^{N_{2}}\lambda_{i,Q_{2}}+P_{c}} \end{array}$$
(20b)$$\begin{array}{cc} \; & s.t.\;\eta \left(1-\alpha \right)\left(\sum_{i=1}^{N_{1}}\lambda_{i,H}\lambda_{i,Q_{1}}+\sigma_{2}^{2}\right)\geq \sum_{i=1}^{N_{2}}\lambda_{i,Q_{2}}+P_{c2} \end{array}$$
(20c)$$\begin{array}{cc} \; & \sum_{i=1}^{N_{1}}\lambda_{i,Q_{1}}\leq {\bar{P}}_{\max}^{1} \end{array}$$
(20d)$$\begin{array}{cc} \; & \sum_{i=1}^{N_{2}}\lambda_{i,Q_{2}}\leq {\bar{P}}_{\max}^{2} \end{array}$$
(20e)$$\begin{array}{cc} \; & \sum_{i=1}^{N_{1}}\log_{2}\left(1+\frac{\alpha \lambda_{i,H}\lambda_{i,Q_{1}}}{\alpha \sigma_{2}^{2}+\sigma_{ID}^{2}}\right)\geq {\bar{r}}_{\min}^{2} \end{array}$$
(20f)$$\begin{array}{cc} \; & \sum_{i=1}^{N_{2}}\log_{2}\left(1+\frac{\lambda_{i,H}\lambda_{i,Q_{2}}}{\sigma_{1}^{2}}\right)\geq {\bar{r}}_{\min}^{1} \end{array}$$
(20g)$$\begin{array}{cc} \; & 0\leq \alpha \leq 1,\lambda_{Q_{1}}\geq 0,\lambda_{Q_{2}}\geq 0, \end{array}$$
where $\lambda_{Q_{1}}$ and $\lambda_{Q_{2}}$ are the set of eigenvalues for $Q_{1}$ and $Q_{2}$, respectively, i.e., $\lambda_{Q_{1}}=\left\{\left.\lambda_{i,Q_{1}}\right|i=1,2,\ldots N_{1}\right\}$, $\lambda_{Q_{2}}=\left\{\left.\lambda_{i,Q_{2}}\right|i=1,2,\ldots N_{2}\right\}$.

Due to the power splitting factor α and the eigenvalues $\lambda_{i,Q_{1}}$ are coupled with each other in both the objective function and the constraint conditions of Equations (20b) and (20e), problem P4 is still a non-convex optimization problem. However, problem P4 can be further divided into two correlated sub-problems and their optimal solutions can still converge to the global optimum in the sequel.

Giving a function $f\left(x\right)=\frac{g_{1}\left(x\right)}{g_{2}\left(x\right)}$, if $g_{1}\left(x\right)$ is a concave function and $g_{2}\left(x\right)$ is an affine function, then f(x) is defined as a pseudo-concave function. As to the objective function of problem P4, as shown in Equation (20a), its numerator can be regarded as a concave function with respect to $\lambda_{Q_{1}}$ and $\lambda_{Q_{2}}$, while its denominator is an affine function of $\lambda_{Q_{1}}$ and $\lambda_{Q_{2}}$, the objective function is hence a pseudo-concave function with respect to variables $\lambda_{Q_{1}}$ and $\lambda_{Q_{2}}$. To verify the concavity, the Hessian matrix in terms of the variable α is given as in Equation (21), where the parameters in Equation (21) are all non-negative. The second derivative of α is negative and thus this validates that the objective function Equation (20a) is also a concave function with respect to α.
(21)$$\begin{array}{c} \nabla_{\alpha }^{2}=-\frac{\sigma_{ID}^{2}}{\ln2}\sum_{i=1}^{N_{1}}\frac{2\alpha \sigma_{2}^{4}+2\sigma_{2}^{2}\sigma_{ID}^{2}+2\alpha \sigma_{2}^{2}\lambda_{i,H}\lambda_{i,Q_{1}}+\sigma_{ID}^{2}\lambda_{i,H}\lambda_{i,Q_{1}}}{{\left(\alpha^{2}\sigma_{2}^{4}+\sigma_{ID}^{4}+2\alpha \sigma_{2}^{2}\sigma_{ID}^{2}+\alpha^{2}\sigma_{2}^{2}\lambda_{i,H}\lambda_{i,Q_{1}}+\alpha \sigma_{ID}^{2}\lambda_{i,H}\lambda_{i,Q_{1}}\right)}^{2}}<0. \end{array}$$

The concavity of the objective function about α indicates that there must be a closed-form solution for α. For each fixed α, there is always a set of feasible solutions for $\left\{\lambda_{Q_{1}},\lambda_{Q_{2}}\right\}$. The optimal eigenvalues of the two precoding matrices will be obtained by calculating the closed-form solution through circular iteration. The analysis and resolving solutions are summarized as follows:


*(1) Optimal $\left\{\lambda_{Q_{1}}^{*},\lambda_{Q_{2}}^{*}\right\}$ for Given α*


For a given α, problem P4 with respect to $\left\{\lambda_{Q_{1}},\lambda_{Q_{2}}\right\}$ can be reduced to P5: (22a)P5:maxλQ1,λQ2Ξ(λQ1,λQ2)Ξ(λQ1,λQ2)Ψ(λQ1,λQ2)Ψ(λQ1,λQ2)(22b)$$\begin{array}{cc} \; & s.t.\;\eta \left(1-\alpha \right)\left(\sum_{i=1}^{N_{1}}\lambda_{i,H}\lambda_{i,Q_{1}}+\sigma_{2}^{2}\right)\geq \sum_{i=1}^{N_{2}}\lambda_{i,Q_{2}}+P_{c2} \end{array}$$(22c)$$\begin{array}{cc} \; & \sum_{i=1}^{N_{1}}\log_{2}\left(1+\frac{\alpha \lambda_{i,H}\lambda_{i,Q_{1}}}{\alpha \sigma_{2}^{2}+\sigma_{ID}^{2}}\right)\geq {\bar{r}}_{\min}^{2} \end{array}$$(22d)$$\begin{array}{cc} \; & \lambda_{Q_{1}}\geq 0,\lambda_{Q_{2}}\geq 0 \end{array}$$(22e)$$\begin{array}{cc} \; & (20c)(20d)(20f), \end{array}$$
where $\Xi \left(\lambda_{Q_{1}},\lambda_{Q_{2}}\right)\;\;\;=\;\;\;\sum_{i=1}^{N_{1}}\log_{2}\left(1+\frac{\alpha \lambda_{i,H}\lambda_{i,Q_{1}}}{\alpha \sigma_{2}^{2}+\sigma_{ID}^{2}}\right)+\sum_{i=1}^{N_{2}}\log_{2}\left(1+\frac{\lambda_{i,H}\lambda_{i,Q_{2}}}{\sigma_{1}^{2}}\right)$ and $\Psi \left(\lambda_{Q_{1}},\lambda_{Q_{2}}\right)=\sum_{i=1}^{N_{1}}\lambda_{i,Q_{1}}+\sum_{i=1}^{N_{2}}\lambda_{i,Q_{2}}+P_{c}$. The constraint of Equation (22b) is affine with respect to $\left\{\lambda_{Q_{1}},\lambda_{Q_{2}}\right\}$, while the constraint of Equation (22e) can be seen as a concave set of $\left\{\lambda_{Q_{1}},\lambda_{Q_{2}}\right\}$. Meanwhile, the objective function of P5 is a non-convex fractional form. Therefore, the Dinkelbach method [31] and CVX can be applied to solve this fractional programming problem P5.

The Dinkelbach method is considered to be an effective iterative algorithm to solve the fractional programming problem with multi-parameters. The Dinkelbach algorithm can convert the objective function from its fractional form to the subtractive form and can guarantee the convergence to the global optimal solution. Herein, the optimal energy efficiency can be redefined as $q^{*}$:(23)q*=Ξ(λQ1*,λQ2*)Ξ(λQ1*,λQ2*)Ψ(λQ1*,λQ2*)Ψ(λQ1*,λQ2*)=max{λQ1,λQ2}∈FΞ(λQ1,λQ2)Ξ(λQ1,λQ2)Ψ(λQ1,λQ2)Ψ(λQ1,λQ2).

In Equation (23), F is a compact feasible set, with $F=\left\{\left.\left(\lambda_{Q_{1}},\lambda_{Q_{2}}\right)\right|\left(22b\right)\left(22c\right)\left(22d\right)\left(22e\right)\right\}$ and $\left\{\lambda_{Q_{1}}^{*},\lambda_{Q_{2}}^{*}\right\}$ represents the optimal value of $\left\{\lambda_{Q_{1}},\lambda_{Q_{2}}\right\}$ to P5. Following the equivalent transformation shown in Equation (23), Theorem 2 can be given as follows.

**Theorem** **2.**
*The maximum energy efficiency $q^{*}$ can be achieved if and only if*
(24)$$\begin{array}{c} \max_{\{\lambda_{Q_{1}},\lambda_{Q_{2}}\}\in F}\Xi \left(\lambda_{Q_{1}},\lambda_{Q_{2}}\right)-q^{*}\Psi \left(\lambda_{Q_{1}},\lambda_{Q_{2}}\right){}_{}^{}{}_{}^{}{}_{}^{}=\Xi \left(\lambda_{Q_{1}}^{*},\lambda_{Q_{2}}^{*}\right)-q^{*}\Psi \left(\lambda_{Q_{1}}^{*},\lambda_{Q_{2}}^{*}\right)=0 \end{array}$$
*with $\Xi (\lambda_{Q_{1}},\lambda_{Q_{2}})\geq 0$, $\Psi (\lambda_{Q_{1}},\lambda_{Q_{2}})>0$.*


**Proof.** Assume that $\left\{\lambda_{Q_{1}}^{*},\lambda_{Q_{2}}^{*}\right\}\in F$ is the optimal precoding strategy for problem P5 and $q^{*}$ is the optimal energy efficiency. For any feasible solutions, they satisfy the following inequality, Equation (25),
(25)$$\begin{array}{c} \frac{\Xi (\lambda_{Q_{1}},\lambda_{Q_{2}})}{\Psi (\lambda_{Q_{1}},\lambda_{Q_{2}})}\leq q^{*}=\frac{\Xi (\lambda_{Q_{1}}^{*},\lambda_{Q_{2}}^{*})}{\Psi (\lambda_{Q_{1}}^{*},\lambda_{Q_{2}}^{*})}. \end{array}$$Equation (25) can be transformed to the following two inequalities, Equations (26) and (27),
(26)$$\begin{array}{cc} \; & \Xi \left(\lambda_{Q_{1}},\lambda_{Q_{2}}\right)-q^{*}\Psi \left(\lambda_{Q_{1}},\lambda_{Q_{2}}\right)\leq 0, \end{array}$$
(27)$$\begin{array}{cc} \; & \Xi \left(\lambda_{Q_{1}}^{*},\lambda_{Q_{2}}^{*}\right)-q^{*}\Psi \left(\lambda_{Q_{1}}^{*},\lambda_{Q_{2}}^{*}\right)=0. \end{array}$$Equations (26) and (27) imply that maximizing the energy efficiency $q^{*}$ is equivalent to finding the optimization variables $\left\{\lambda_{Q_{1}}^{*},\lambda_{Q_{2}}^{*}\right\}$ that satisfy $\max_{\{\lambda_{Q_{1}},\lambda_{Q_{2}}\}\in F}\Xi \left(\lambda_{Q_{1}},\lambda_{Q_{2}}\right)-q^{*}\Psi \left(\lambda_{Q_{1}},\lambda_{Q_{2}}\right)=0$. □

Theorem 2 validates the theoretical operability for the Dinkelbach method. It enables the fractional programming problem to be transformed to the subtractive-form via introducing *q*. The optimal solution of the fractional programming problem can be indirectly obtained by solving the subtractive-form optimization problem. The problem P6, which is converted from problem P5, is hence obtained and shown as follows
(28)$$\begin{array}{c} P6:\max_{\{\lambda_{Q_{1}},\lambda_{Q_{2}}\}\in F}\Xi \left(\lambda_{Q_{1}},\lambda_{Q_{2}}\right)-q\Psi \left(\lambda_{Q_{1}},\lambda_{Q_{2}}\right). \end{array}$$

Based on the Dinkelbach method, *q* can be iterated from an initial point and the convergence condition proposed in Theorem 2 should be verified in each iteration. In other words, substituting *q* of each iteration into P6 verifies whether the objective function of P6 can be converged to the maximum tolerance. If the algorithm converges, the optimal solution is returned; otherwise, *q* is updated in the following iteration until the maximum tolerance is satisfied.


*(2) Optimal $\alpha^{*}$ for Given $\left\{\lambda_{Q_{1}},\lambda_{Q_{2}}\right\}$*


Given a feasible $\left\{\lambda_{Q_{1}},\lambda_{Q_{2}}\right\}$, problem P4 with respect to α can be reduced to P7
(29a)$$\begin{array}{cc} P7: & \max_{\alpha }\overset{}{\underset{}{}}\frac{\sum_{i=1}^{N_{1}}\log_{2}\left(1+\frac{\alpha \lambda_{i,H}\lambda_{i,Q_{1}}}{\alpha \sigma_{2}^{2}+\sigma_{ID}^{2}}\right)+\sum_{i=1}^{N_{2}}\log_{2}\left(1+\frac{\lambda_{i,H}\lambda_{i,Q_{2}}}{\sigma_{1}^{2}}\right)}{\sum_{i=1}^{N_{1}}\lambda_{i,Q_{1}}+\sum_{i=1}^{N_{2}}\lambda_{i,Q_{2}}+P_{c}} \end{array}$$
(29b)$$\begin{array}{cc} \; & s.t.\;\eta \left(1-\alpha \right)\left(\sum_{i=1}^{N_{1}}\lambda_{i,H}\lambda_{i,Q_{1}}+\sigma_{2}^{2}\right)\geq \sum_{i=1}^{N_{2}}\lambda_{i,Q_{2}}+P_{c2} \end{array}$$
(29c)$$\begin{array}{cc} \; & \sum_{i=1}^{N_{1}}\log_{2}\left(1+\frac{\alpha \lambda_{i,H}\lambda_{i,Q_{1}}}{\alpha \sigma_{2}^{2}+\sigma_{ID}^{2}}\right)\geq {\bar{r}}_{\min}^{2} \end{array}$$
(29d)$$\begin{array}{cc} \; & 0\leq \alpha \leq 1. \end{array}$$

Due to the objective function of P7 being concave with respect to α, the optimization problem P7 has a closed-form solution for it. When taking the EE function’s first derivative about α, the inequality $\frac{\partial \mathrm{EE}}{\partial \alpha }>0$ is obtained. This means that EE is a monotonically increasing function with respect to α. Then, the constraint of Equation (29b) can be processed by equivalent transformation and Equation (30) can be obtained as follows
(30)$$\begin{array}{c} \alpha \leq 1-\frac{\sum_{i=1}^{N_{2}}\lambda_{i,Q_{2}}+P_{c2}}{\eta (\sum_{i=1}^{N_{1}}\lambda_{i,H}\lambda_{i,Q_{1}}+\sigma_{2}^{2})}. \end{array}$$

The optimal variable alpha in P7 must achieve the upper bound $\left(1-\frac{\sum_{i=1}^{N_{2}}\lambda_{i,Q_{2}}+P_{c2}}{\eta (\sum_{i=1}^{N_{1}}\lambda_{i,H}\lambda_{i,Q_{1}}+\sigma_{2}^{2})}\right)$ constrained by Equation (30). Otherwise, the objective function of P7 can obtain a larger value via increasing α instead of violating any constraints. Thus, if the optimization variables $\left\{\lambda_{Q_{1}},\lambda_{Q_{2}}\right\}$ are maintained as constants, the optimal α can be obtained as
(31)$$\begin{array}{c} \alpha^{*}=1-\frac{\sum_{i=1}^{N_{2}}\lambda_{i,Q_{2}}+P_{c2}}{\eta (\sum_{i=1}^{N_{1}}\lambda_{i,H}\lambda_{i,Q_{1}}+\sigma_{2}^{2})}. \end{array}$$

**Remark** **1.**
*In Equation (31), all parameters in the second term on the right side of the equal sign are positive. This means that the optimal value of α is in $\left[0,1\right)$. The optimization variable α cannot be negative or the problem P7 will be infeasible.*



*(3) Joint optimization for $\left\{\lambda_{Q_{1}},\lambda_{Q_{2}}\right\}$ and α*


With the Dinkelbach method, $\left\{\lambda_{Q_{1}},\lambda_{Q_{2}}\right\}$ can converge to their optimums by fixing α. Consequently, an alternating algorithm based on the Dinkelbach method can also be proposed to guarantee that both $\left\{\lambda_{Q_{1}}^{*},\lambda_{Q_{2}}^{*}\right\}$ and $\alpha^{*}$ converge to their optimums synchronously. The procedures of the proposed alternating algorithm based on the Dinkelbach method are explained using the pseudo-code. Here, *m* and *n* denote the iteration numbers of α and $\left\{\lambda_{Q_{1}},\lambda_{Q_{2}}\right\}$, respectively.

In the pseudo-code of the proposed algorithm (Algorithm 1), with an initialized $\alpha^{\left(m=0\right)}$, the Dinkelbach method is adopted in order to find $\left\{\lambda_{Q_{1}}^{\left(n\right)},\lambda_{Q_{2}}^{\left(n\right)}\right\}$. Then, with the known $\left\{\lambda_{Q_{1}}^{\left(n\right)},\lambda_{Q_{2}}^{\left(n\right)}\right\}$, $\alpha^{\left(m+1\right)}$ is updated according to Equation (31). The iterations will terminate until the convergence tolerance is met and then the optimal variables can be obtained.

**Algorithm 1:** Alternating algorithm based on Dinkelbach1. **Initialization**
 **Set** maximum tolerance ε1, ε2 and iterations n=0, m=0; **Set** initial points α(0), q=0 and qpre=0;2. **Repeat**3.  **Repeat**4.   Substitute α(m) into Equation (28) and use CVX to calculate λQ1(n),λQ2(n);5.   **If**
Ξ(α(m),λQ1(n),λQ2(n))−qΨ(λQ1(n),λQ2(n))<ε16.    Convergence = true;7.    **Return**
(α*,λQ1*,λQ2*)=(α(m),λQ1(n),λQ2(n)) and q*=Ξ(α*,λQ1*,λQ2*)Ξ(α*,λQ1*,λQ2*)Ψ(λQ1*,λQ2*)Ψ(λQ1*,λQ2*);8.   **Else**
q=Ξ(α(m),λQ1(n),λQ2(n))Ξ(α(m),λQ1(n),λQ2(n))Ψ(λQ1(n),λQ2(n))Ψ(λQ1(n),λQ2(n)) and n=n+1;9.    Convergence = false;10.  **End**11. **Until** Convergence = true12.  **If**
q*−qpre≤ε213.    Convergence = true;14.  **Else** Calculated α(m+1) according to Equation (31), m=m+1 and α(m)=α(m+1);15.   qpre=q*;16.   Convergence = false;17.  **End**18. **Until** Convergence = true

***Convergence and Optimality Analysis:***The objective function of P4 is joint pseudo-concave with $\left\{\lambda_{Q_{1}},\lambda_{Q_{2}}\right\}$ for each fixed α. By observing the compact feasible set $F=\left\{\left.\left(\lambda_{Q_{1}},\lambda_{Q_{2}}\right)\right|\left(22b\right)\left(22c\right)\left(22d\right)\left(22e\right)\right\}$, it can conclude that there is no coupling between the optimized variables. Therefore, the set F is a strictly convex feasible set and can ensure the convergence to a Karush–Kuhn–Tucker (KKT) point [32]. Meanwhile, according to the Dinkelbach method, we transform the objective function of P5 to the subtractive form in P6 and combine P6 with the convex feasible set F to obtain a convex optimization problem. So we can get the optimal value of $\left\{\lambda_{Q_{1}}^{*},\lambda_{Q_{2}}^{*}\right\}$ by the internal loop of the proposed algorithm.

In the outer loop, the proposed algorithm generates an increasing objective function at each iteration, which means the optimization problem is upper-bounded and hence the convergence of P7 is guaranteed. According to the derivative that $\frac{\partial \mathrm{EE}}{\partial \alpha }>0$, it can be concluded that, when fixing the other variables except α, the objective function of P7 is a monotonically increasing function with respect to α. In the proposed algorithm, for a given $\left\{\lambda_{Q_{1}}^{*},\lambda_{Q_{2}}^{*}\right\}$, α is optimized to increase EE, which means that the value of α will increase with iterations. Therefore, if α is initialized to be extremely small, the proposed algorithm will converge.

***Complexity Analysis:*** Now we will analyze the computational complexity of the proposed alternating algorithm based on Dinkelbach. The computational complexity is inferred based on the optimization problem P4. It must be known that the Dinkelbach algorithm has the characteristic of super-linear convergence with $K_{1}$ iterations. In addition, the computational complexity can be represented by a polynomial composed of the number of variables and constraints contained in the algorithm’s internal loop due to the optimization problem P4 being convex, i.e., $O(K_{1}\left(N_{1}+N_{2}\right))$. The computational complexity for searching α is O(1). Thus, the total complexity of the proposed algorithm is $O(K_{2}\left(K_{1}\left(N_{1}+N_{2}\right)+1\right))$, where $K_{2}$ is the number of iterations within the outer loop.

## 4. The Multiple EH Sensor Nodes Scenario

The energy efficiency maximization problem in the extended scenario with one source and *K* EH sensor nodes being used is considered. Similar to the settings in the single node scenario, the source is equipped with $N_{1}$ antennas and each sensor node is equipped with $N_{2}$ antennas.

The received signals at the *k*-th EH sensor node can be given as in Equation (32),
(32)$$\begin{array}{cc} y_{2,k} & =\sum_{k=1}^{K}H_{k}F_{k}\vartheta_{1,k}+n_{2,k}\\ \; & =\underset{Desired\;signal}{\underset{︸}{H_{k}F_{k}\vartheta_{1,k}}}+\underset{Interference\;signal}{\underset{︸}{\sum_{l\neq k}^{K}H_{k}F_{l}\vartheta_{1,l}}}+\underset{AWGN}{\underset{︸}{n_{2,k}}}, \end{array}$$
where $\vartheta_{1,k}\in C^{N_{1}\times 1}$ is the symbol vector for the *k*-th EH sensor node. In particular, it can be assumed that EH sensor nodes can remove all interference signals before decoding the information of a desired signal [33]. Thus, the achievable sum-rate in bits/s/Hz of *K* EH sensor nodes can be given as in Equation (33),
(33)$$\begin{array}{c} R_{2,sensors}=\sum_{k=1}^{K}\log_{2}\left|I_{N_{2}}+\frac{\alpha H_{k}Q_{1,k}H_{k}^{H}}{\left(\alpha \sigma_{2,k}^{2}+\sigma_{ID,k}^{2}\right)I_{N_{2}}}\right|, \end{array}$$
where $Q_{1,k}=F_{k}F_{k}^{H}$. Meanwhile, the amount of totally harvested energy of *K* EH sensor nodes is shown as follows
(34)$$\begin{array}{c} E_{EH,\mathrm{tol}}=\eta \left(1-\alpha \right)Tr\left(\sum_{k=1}^{K}(H_{k}Q_{1,k}H_{k}^{H}+\sigma_{2,k}^{2}I_{N_{2}})\right). \end{array}$$

When the sensor nodes transmit information to the source by utilizing the harvested energy, the received signals at the source can be deduced as in Equation (35),
(35)$$\begin{array}{c} y_{1,source}=\sum_{k=1}^{K}H_{k}^{H}D_{k}\vartheta_{2,k}+n_{1}. \end{array}$$

According to Equation (35), the achievable data rate of the source is given by Equation (36),
(36)$$\begin{array}{c} R_{1,source}=\log_{2}\left|I_{N_{1}}+\frac{\sum_{k=1}^{K}H_{k}^{H}Q_{2,k}H_{k}}{\sigma_{1}^{2}I_{N_{1}}}\right|, \end{array}$$
where $Q_{2,k}=D_{k}D_{k}^{H}$.

Thus, the energy efficiency of the MIMO half-duplex wireless sensor networks with multiple EH sensor nodes can be denoted as
(37)$$\begin{array}{c} EE_{mul}=\frac{R_{2,sensors}+R_{1,source}}{P_{\mathrm{tol}}}, \end{array}$$
where $P_{\mathrm{tol}}={\tilde{P}}_{1}+{\tilde{P}}_{2}+{\tilde{P}}_{c}$, ${\tilde{P}}_{1}=\sum_{k=1}^{K}Tr(Q_{1,k})$, ${\tilde{P}}_{2}=\sum_{k=1}^{K}Tr(Q_{2,k})$ and ${\tilde{P}}_{c}=P_{c1}+KP_{c2}$. The circuit power consumption of an EH sensor node is assumed to be the same for each other and hence $KP_{c2}$ denotes the total circuit power consumption of all sensor nodes.

For the formulation of the energy efficiency maximization problem, this paper adopts the same constraints as initially considered in the single node scenario. Therefore, the optimization problem can be formulated as **P8**,
(38a)$$\begin{array}{cc} P8: & \max_{\alpha ,Q_{1,k},Q_{2,k}}\frac{R_{2,sensors}+R_{1,source}}{P_{\mathrm{tol}}} \end{array}$$
(38b)$$\begin{array}{cc} \; & s.t.E_{EH,\mathrm{tol}}\geq {\tilde{P}}_{2}+KP_{c2} \end{array}$$
(38c)$$\begin{array}{cc} \; & \sum_{k=1}^{K}Tr(Q_{1,k})\leq {\tilde{P}}_{\max}^{1} \end{array}$$
(38d)$$\begin{array}{cc} \; & \sum_{k=1}^{K}Tr(Q_{2,k})\leq {\tilde{P}}_{\max}^{2} \end{array}$$
(38e)$$\begin{array}{cc} \; & R_{2,sensors}\geq {\tilde{r}}_{\min}^{2} \end{array}$$
(38f)$$\begin{array}{cc} \; & R_{1,source}\geq {\tilde{r}}_{\min}^{1} \end{array}$$
(38g)$$\begin{array}{cc} \; & 0\leq \alpha \leq 1,Q_{1,k}⪰0,Q_{2,k}⪰0, \end{array}$$
where Equation (38b) is the harvested energy constraint, ${\tilde{P}}_{\max}^{1}$ is the upper limit of the source’s transmitting power and ${\tilde{P}}_{\max}^{2}$ is the total available power of all sensor nodes. ${\tilde{r}}_{\min}^{2}$ and ${\tilde{r}}_{\min}^{1}$ are the required rate threshold of *K* nodes and the source, respectively.

By applying the proposed optimal solution in Section 3.2 to problem **P8**, the optimization problem can be simplified as **P9**.
(39a)$$\begin{array}{cc} P9: & \max_{\alpha ,\lambda_{Q_{1,k}},\lambda_{Q_{2,k}}}\frac{\sum_{k=1}^{K}\sum_{i=1}^{N_{1}}\log_{2}\left(1+\frac{\alpha \lambda_{i,H_{k}}\lambda_{i,Q_{1,k}}}{\alpha \sigma_{2,k}^{2}+\sigma_{ID,k}^{2}}\right)+\sum_{i=1}^{N_{2}}\log_{2}\left(1+\frac{\sum_{k=1}^{K}\lambda_{i,H_{k}}\lambda_{i,Q_{2,k}}}{\sigma_{1}^{2}}\right)}{\sum_{k=1}^{K}\sum_{i=1}^{N_{1}}\lambda_{i,Q_{1,k}}+\sum_{k=1}^{K}\sum_{i=1}^{N_{2}}\lambda_{i,Q_{2,k}}+{\tilde{P}}_{c}} \end{array}$$
(39b)$$\begin{array}{cc} \; & s.t.\eta \left(1-\alpha \right)\sum_{k=1}^{K}\left(\sum_{i=1}^{N_{1}}\lambda_{i,H_{k}}\lambda_{i,Q_{1,k}}+\sigma_{2,k}^{2}\right)\geq \sum_{k=1}^{K}\sum_{i=1}^{N_{2}}\lambda_{i,Q_{2,k}}+KP_{c2} \end{array}$$
(39c)$$\begin{array}{cc} \; & \sum_{k=1}^{K}\sum_{i=1}^{N_{1}}\lambda_{i,Q_{1,k}}\leq {\tilde{P}}_{\max}^{1} \end{array}$$
(39d)$$\begin{array}{cc} \; & \sum_{k=1}^{K}\sum_{i=1}^{N_{2}}\lambda_{i,Q_{2,k}}\leq {\tilde{P}}_{\max}^{2} \end{array}$$
(39e)$$\begin{array}{cc} \; & \sum_{k=1}^{K}\sum_{i=1}^{N_{1}}\log_{2}\left(1+\frac{\alpha \lambda_{i,H_{k}}\lambda_{i,Q_{1,k}}}{\alpha \sigma_{2,k}^{2}+\sigma_{ID,k}^{2}}\right)\geq {\tilde{r}}_{\min}^{2} \end{array}$$
(39f)$$\begin{array}{cc} \; & \sum_{i=1}^{N_{2}}\log_{2}\left(1+\frac{\sum_{k=1}^{K}\lambda_{i,H_{k}}\lambda_{i,Q_{2,k}}}{\sigma_{1}^{2}}\right)\geq {\tilde{r}}_{\min}^{1} \end{array}$$
(39g)$$\begin{array}{cc} \; & 0\leq \alpha \leq 1,\lambda_{Q_{1,k}}\geq 0,\lambda_{Q_{2,k}}\geq 0, \end{array}$$
where $\lambda_{Q_{1,k}}$ and $\lambda_{Q_{2,k}}$ are the set of eigenvalues of $Q_{1,k}$ and $Q_{2,k}$, respectively, i.e., $\lambda_{Q_{1,k}}=\left\{\lambda_{i,Q_{1,k}}\left|i=1,2,\ldots N_{1},\forall k\right.\right\}$ and $\lambda_{Q_{2,k}}=\left\{\lambda_{i,Q_{2,k}}\left|i=1,2,\ldots N_{2},\forall k\right.\right\}$.

The problem **P9** can be solved by the alternating algorithm based on Dinkelbach, and the simulation results are described in Section 5.2.

## 5. Simulation Results

### 5.1. Single EH Sensor Node Scenario

In this subsection, numerical results are exhibited to show the performances of the proposed alternating algorithm based on the Dinkelbach method. In the simulations, an EE maximization precoding design in a SWIPT-based MIMO half-duplex wireless sensor network is presented by setting $N_{1}=N_{2}$, i.e., the number of the equipped antennas at the source is equal to that of the sensor node. The block fading channels modeled as complex Gaussian random variables are adopted in this paper. The elements of the channel coefficient matrix H are generated complying with the distribution CN(0,1). The maximal transmitting power and the noise power at the source and the sensor node are assumed to be the same, i.e., ${\bar{P}}_{\max}^{1}={\bar{P}}_{\max}^{2}=P_{\max}$ and $\sigma_{1}^{2}=\sigma_{2}^{2}=\sigma_{ID}^{2}=\sigma^{2}=0.2$[15]. The total circuit power consumption can be set as $P_{c}=3W$. The source and the sensor node have the same requirement on the data rate threshold, i.e., ${\bar{r}}_{\min}^{1}={\bar{r}}_{\min}^{2}=r_{\min}$. The energy conversion efficiency is simplified as η=1[10]. In other simulating scenarios, as different η is used, the alternating algorithm maintains the same procedures just as that of η=1. All the numerical simulation results are averaged over 500 independent randomly generated channel realizations unless otherwise specified.
$$\begin{array}{c} H^{\prime }=\left[\begin{array}{cccc} -0.2772+1.1585i & -0.2124-0.2818i & -0.4562-0.0982i & -0.1645+0.4383i\\ 0.4567-0.4329i & 0.3644-0.4480i & 0.0737+1.1462i & -1.3267+0.4853i\\ 1.4030-0.0115i & 0.7624-0.7654i & -0.5862+0.4177i & 0.0699-0.2621i\\ -0.2058-0.1429i & 0.3878+0.6187i & 0.1906+0.1061i & 0.4486-1.6408i \end{array}\right] \end{array}$$

Figure 2 presents the energy efficiency achieved by the proposed algorithm versus the number of iterations. The results are obtained using five different initial points by setting the maximal transmitting power $P_{\max}=\;8$ W and the data rate threshold $r_{\min}=1$ bps/Hz. The channel coefficient matrix $H^{\prime }$ is generated complying with the distribution CN(0,1). The source and the sensor node are equipped with the same amount of antennas by setting $N_{1}=N_{2}=4$. Simulation results in Figure 2 validate that the proposed algorithm enables the energy efficiency to converge to its optimum value 2.7504 bits/Hz/J with no more than three iteration times when iterating from five different points.

In Figure 3, the performance comparisons among the alternating algorithm and the other two baseline schemes, i.e., the non-EH restriction scheme and the non-precoding design for the source scheme, are conducted by setting the other parameters as $N_{1}=N_{2}=4$ and $r_{\min}=1$ bps/Hz. The non-EH restriction scheme means that both the source and the sensor node are served with the precoding design and are supplied with sufficient energy. It is unnecessary for the sensor node to harvest the RF energy from the source. The received signals at the sensor node are decoded as the data information. In addition, the non-precoding design for the source scheme means that each diagonal element of the source’s eigenvalue matrix is equal to PmaxPmaxN1N1. The simulation results show that the energy efficiency of the non-EH restriction scheme is the highest among all the baseline algorithms. This is because all of the received signals at the sensor node are used to transmit information, then the sensor node can achieve a higher transmission rate at the same energy consumption in the non-EH restriction scheme. However, in the scheme of a non-precoding design for the source, the transmission power is distributed evenly across each antenna of the source. The energy efficiency of this scheme is always the lowest among all baseline algorithms. The energy efficiency without precoding even decreases if $P_{\max}$ continues to improve. This is because the transmitting power of the source cannot be properly configured according to the channel states between antenna pairs, so the information rate of the sensor node is lower at the same energy consumption. By contrast, the energy efficiency in the proposed alternating algorithm based on Dinkelbach can rapidly converge to an optimal value. Unlike the non-precoding design scheme, the energy efficiency will not descend with the increase of $P_{\max}$. To sum up, the precoding design scheme we designed can not only achieve a higher energy efficiency than the non-precoding scheme, but also rapidly converges as $P_{\max}$ increases.

Figure 4 presents the performance comparison between our proposed alternating algorithm and the non-EH restriction scheme by varying the number of antennas at both the source and the sensor node. In the simulation, the required rate is set as $r_{\min}=1$ bps/Hz and the influences on energy efficiency as varying the quantity of antennas are compared by gradually increasing $P_{\max}$ in the two schemes. It can be intuitively observed from Figure 4 that the more antennas are equipped, the higher the energy efficiency obtained in each scheme. This is because the increase in the number of antennas can enhance the spatial freedom degree and the antennas’ diversity gain. Meanwhile, it can be concluded from Figure 4 that the energy efficiency in the proposed “alternating algorithm based on Dinkelbach” scheme approximates to that of the non-EH restriction scheme at two scenarios with a different number of antennas.

In Figure 5, the energy efficiency performance is studied by changing the required minimum data rate. The antenna numbers in subgraphs (a) and (b) are configured as $N_{1}=N_{2}=4$ and $N_{1}=N_{2}=8$, respectively. The transmission power constraints of subgraphs (a) and (b) are all set as $P_{\max}=8W$. It can be observed from Figure 5 that the highest energy efficiencies shown in the two subgraphs are obtained at two distinct required minimum data rates, i.e., 3 bps/Hz and 6 bps/Hz, respectively. As the required data rate is improved, the energy efficiency improves and then it drops with the rate increment. This is because, by lifting the data rate threshold, the actual transmission power of antennas needs to be increased synchronously. According to the objective function of P4 shown as the fraction in Equation (20a), the energy efficiency is simultaneously affected by the joint contribution of its numerator and denominator. The variation of transmission power influences the fraction value. When the energy efficiency achieves its maximum, continuing to increase the data rate threshold will cause the energy efficiency to decrease inversely.

In Figure 6, the probability of feasibility under different $P_{\max}$ settings is investigated by increasing the required minimum data rate. If the achievable data rate fails the rate threshold, the maximization problem P4 will be infeasible, i.e., no optimal solutions to P4 exist. In order to verify the probability feasibility under different data rate thresholds, the number of antennas at transceivers is set as $N_{1}=N_{2}=4$. When the amount of feasible solutions to problem P4 is less than half of the total simulation results, the problem is called ergodically infeasible [34]. The result of Figure 6 shows that the probability of feasibility decreases with the increase of the minimum required data rate. The higher the transmission power $P_{\max}$ that can be obtained, the larger probability of feasibility achieved.

### 5.2. Multiple EH Sensor Nodes Scenario

In this subsection, the energy efficiency performance of the proposed algorithm is evaluated in the networks with multiple EH sensor nodes. In the scenario, three EH sensor nodes, i.e., K=3 are simulated in the scenario and the quantity of the equipped antennas at the source equals that of each sensor node, such that $N_{1}=N_{2}=4$. The maximal transmitting power and the data rate threshold for both the source and each EH sensor node are given as ${\tilde{P}}_{\max}^{1}={\tilde{P}}_{\max}^{2}=P_{\max}$ and ${\tilde{r}}_{\min}^{1}={\tilde{r}}_{\min}^{2}=3\mathrm{bps}/\mathrm{Hz}$, respectively. The other parameters are kept the same as those used in the single EH sensor node scenario.

In Figure 7, the energy efficiency performance is obtained by varying the maximal transmitting power $P_{\max}$. The two baseline algorithms for comparison are designed by assuming each sensor node with indefinite power supply (i.e., non-EH restriction) and by omitting the source precoding design (i.e., non-source precoding), respectively. Distinct from the single EH sensor node scenario, the source eigenvalue matrix in the baseline algorithm of non-source precoding can be obtained as $\frac{P_{\max}}{KN_{1}}I_{N_{1}}$. This is because the source has to allocate power for each sensor node. By observing Figure 7, it can be seen that the energy efficiency of the proposed alternating algorithm based on Dinkelbach outperforms that of the non-precoding design algorithm. As the maximal transmitting power $P_{\max}$ ranges from 1 W to 1.8 W, the energy efficiency of the non-source precoding algorithm falls to zero and the value range of $P_{\max}$ can be deemed as the outage range, which is due to fact that the evenly distributed source power $P_{\max}$ on each antenna cannot support the required sum rate, i.e., the sum rate cannot meet the rate threshold. In contrast, the maximal transmitting power $P_{\max}$ at the source can be optimally assigned to each antenna and hence be capable of ensuring the optimal performance. The energy efficiency of the non-EH restriction algorithm is always the highest among all the candidates, since a much higher achievable data rate at sensor nodes can be obtained when all sensor nodes are free from the constraint of power use.

Figure 8 presents the performance comparison between the alternating algorithm and the fixed power-splitting scheme. In the fixed power-splitting scheme, the power splitting factor α is fixed. In the numerical experiment, the power splitting factor was taken as 0.2, 0.4, 0.6 and 0.8, separately. Figure 8 shows that with the increase of maximal transmitting power, the energy efficiency of all schemes exhibits the monotonous non-decreasing tendency. Meanwhile, it is observed from Figure 8 that the alternating algorithm proposed in the paper can always achieve higher energy efficiency. This is because the fixed power-splitting scheme is unable to guarantee the optimized trade-off of power use between information decoding and energy harvesting, thus resulting in lowering the achievable data rate and consuming more energy. When the maximal transmitting power ranges from 1 W to 2 W, the energy efficiency of the fixed power-splitting scheme falls to 0 as α=0.8. This is because a greater proportion of source power is spent on information decoding and correspondingly the sensor node cannot harvest enough RF energy to satisfy the data rate constraint. The optimization problem P9 is essentially infeasible. In the numerical experiment, the proposed alternating algorithm can always achieve better performance than its counterpart, i.e., a fixed power-splitting scheme.

## 6. Conclusions

In this paper, energy efficiency optimization is investigated in a single-hop MIMO half-duplex wireless sensor network considering a SWIPT scheme. The perfect channel state information is introduced in the energy efficiency maximization problem, where the precoding matrix of the source and the EH sensor node as well as the power splitting factor are jointly designed. The formulated non-convex problem can be equivalently converted by the eigenvalue and singular value decomposition strategy. The alternating algorithm based on Dinkelbach is proposed to solve the converted non-convex optimization problem. The proposed algorithm is validated to achieve the optimality. Simulation results are analyzed to verify the performance of different scenarios for the baseline algorithms. The simulation results evaluate the factors influencing energy efficiency such as the antenna number and data rate threshold. It is shown that the precoding design scheme proposed in the paper can significantly improve energy efficiency. Compared with the non-source precoding scheme, the precoding scheme proposed in the paper shows rapid convergence behavior. These results may provide some insights for better understanding energy efficiency in the multi-antenna SWIPT system. In our future work, the energy efficiency maximization problem will be investigated in the half-duplex SWIPT system with more transceivers.

## Figures and Tables

**Figure 1 sensors-19-04923-f001:**
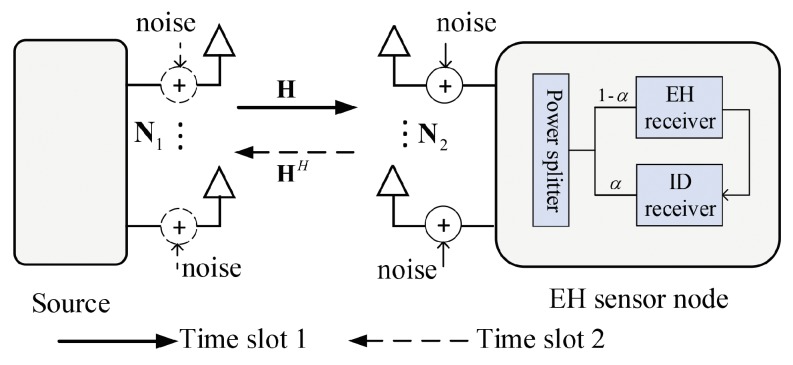
The system model for simultaneous wireless information and power transfer (SWIPT) in a multiple-input multiple-output (MIMO) half-duplex wireless sensor network.

**Figure 2 sensors-19-04923-f002:**
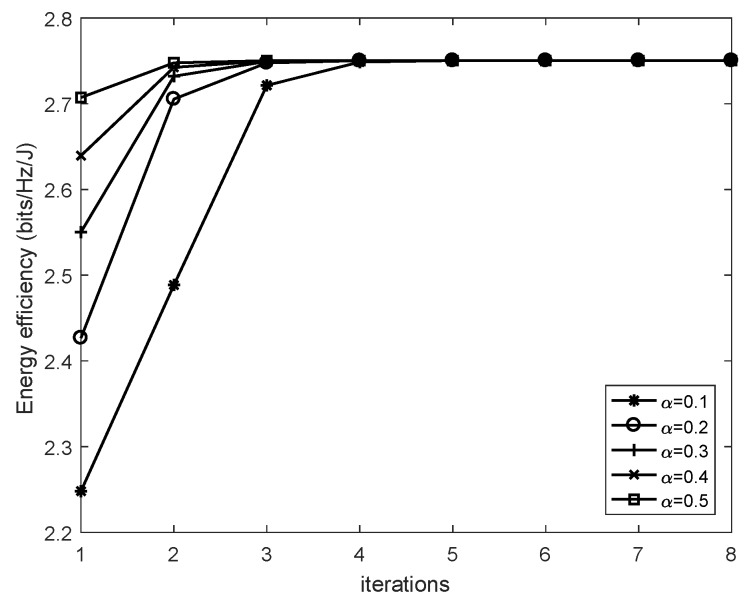
Energy efficiency of the proposed algorithm with five different initial points versus the number of iterations.

**Figure 3 sensors-19-04923-f003:**
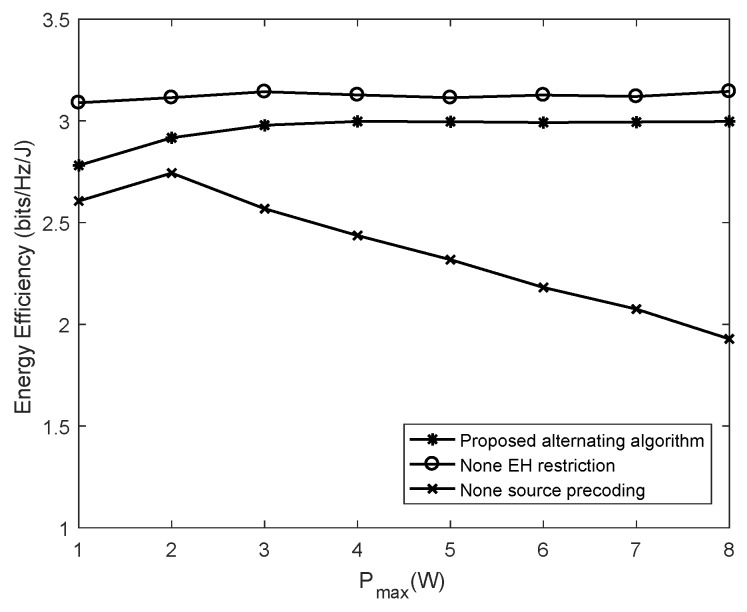
Energy efficiency of the proposed algorithm versus the maximal transmitting power.

**Figure 4 sensors-19-04923-f004:**
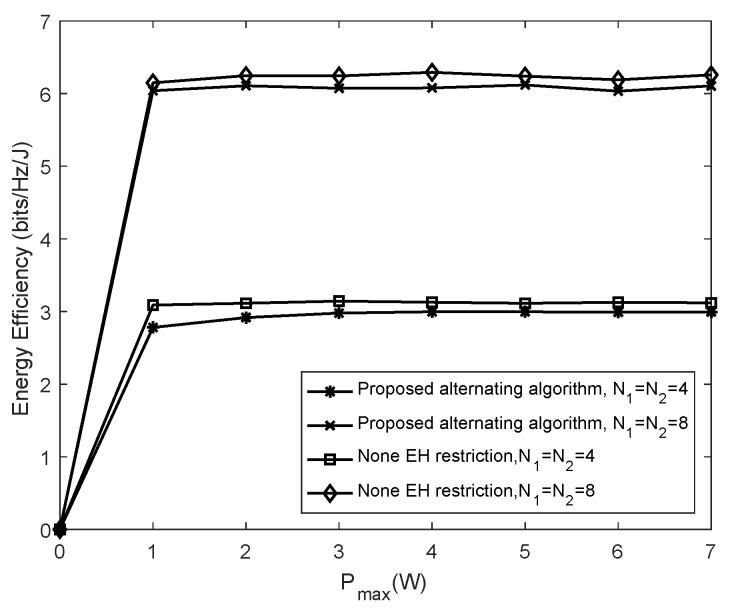
Energy efficiency versus maximum transmitting power when varying the number of antennas.

**Figure 5 sensors-19-04923-f005:**
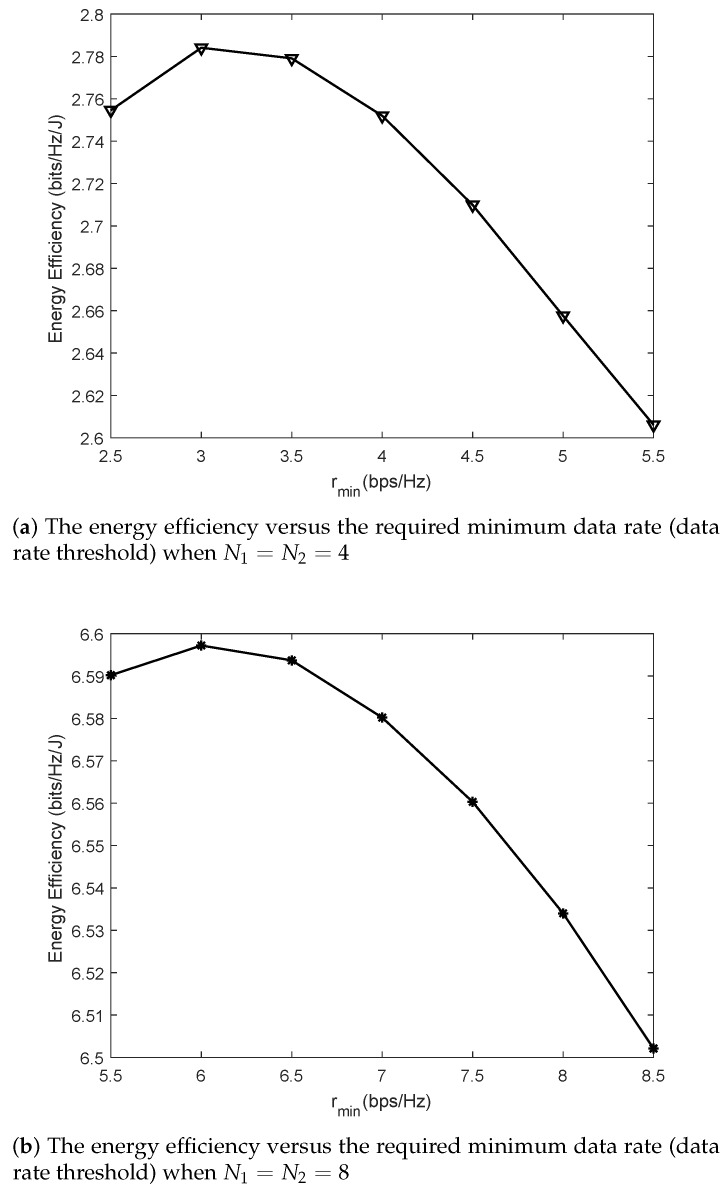
The performance comparison of energy efficiency by varying the required minimum data rate and the quantity of equipped antennas.

**Figure 6 sensors-19-04923-f006:**
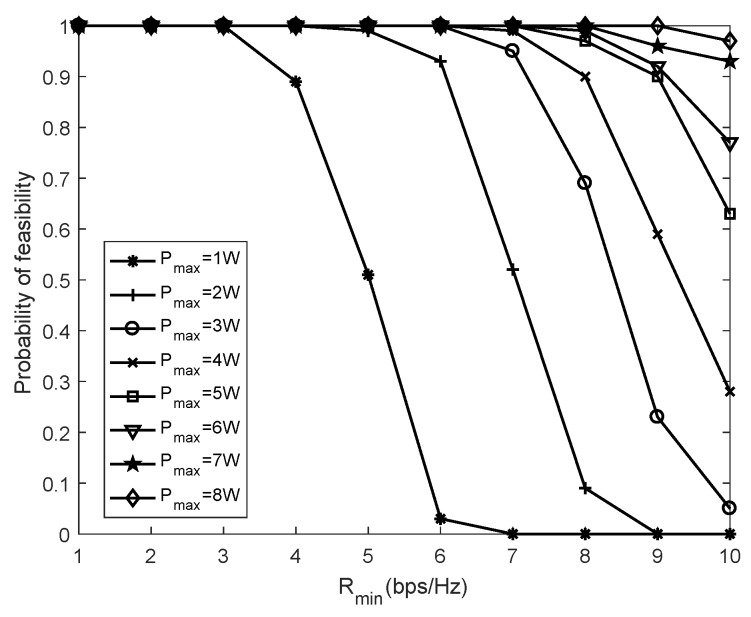
Probability of feasibility versus the minimum data rate requirement.

**Figure 7 sensors-19-04923-f007:**
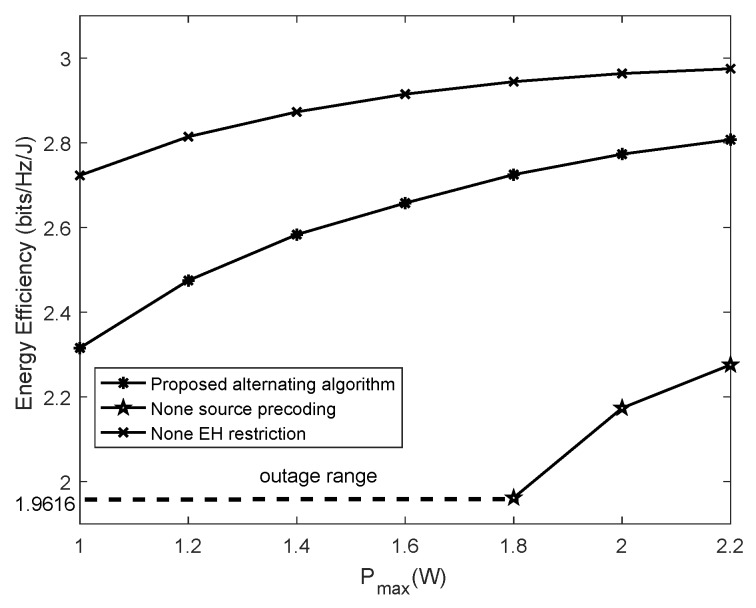
Energy efficiency versus maximal transmitting power in multiple EH sensor nodes scenario.

**Figure 8 sensors-19-04923-f008:**
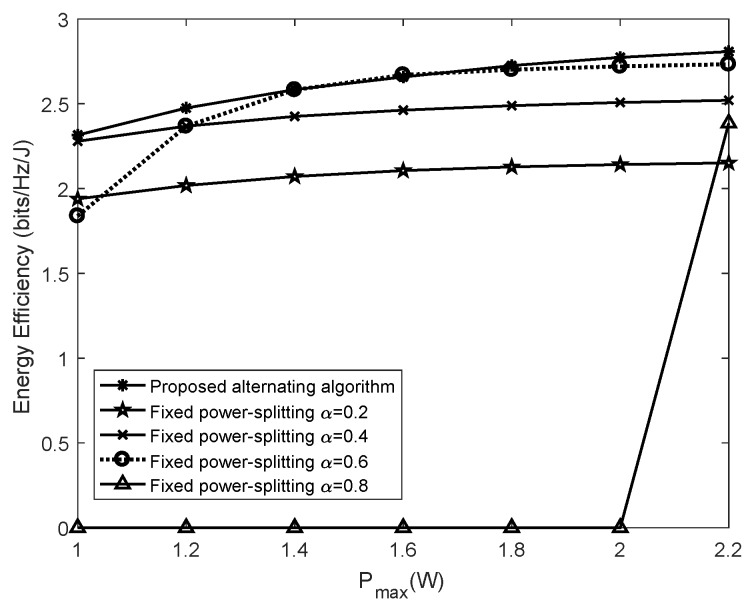
The comparison of energy efficiency between the proposed algorithm and the fixed power-splitting scheme.

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
