# Peer review of "Precoding Design for Energy Efficiency Maximization in MIMO Half-Duplex Wireless Sensor Networks with SWIPT"

_sensors, 2019, doi:10.3390/s19224923_

Round 1

Reviewer 1 Report

Revisions:

1.Authors should investigate  the energy efficiency maximization problem  in the half-duplex SWIPT system with more transceivers.

2.Important references are missing and should be added:

Cheng H, Su Z, Xiong N, et al. Energy-efficient node scheduling algorithms for wireless sensor networks using Markov Random Field model[J]. information sciences, 2016, 329: 461-477. Guo W, Li J, Chen G, et al. A PSO-optimized real-time fault-tolerant task allocation algorithm in wireless sensor networks[J]. IEEE Transactions on Parallel and Distributed Systems, 2015, 26(12): 3236-3249 Cheng H, Xiong N, Yang L T, et al. Distributed scheduling algorithms for channel access in TDMA wireless mesh networks[J]. The Journal of Supercomputing, 2013, 63(2): 407-430

Y.Yao et al:EDAL: An energy-efficient, delay-aware, and lifetime-balancing data collection protocol for heterogeneous wireless sensor networks.IEEE/ACM Transactions on Networking (TON) 23 (3), 810-823, 2015

Reviewer 2 Report

This paper proposes a new algorithm to solve the energy efficiency maximization problem in a single-hop multiple-input multiple-output (MIMO) half-duplex wireless sensor network with simultaneous wireless information and power transfer (SWIPT).Simulation results demonstrate the effectiveness of the proposed algorithm.But, there are still some problems in the formulation of the paper, which need to be revised.

Strong points:

The topic selection of the paper has good practical value. The paper has a complete simulation experiment, which provides sufficient experimental proof for the proposed algorithm.

Weak points:

In the paper, the notations in the introduction should be described as a separate part together with two theorems appearing in the following paper. The fourth part of the paper should be supplemented with a comprehensive experimental analysis to facilitate the reader's research. For example, there are extra square symbols on lines 238 and 270 of the paper, and the formulaon line 256 are not neatly written.

Reviewer 3 Report

This paper aims to maximize the energy efficiency in single-hop MIMO half-duplex SWIPT networks with linear EH model. The power splitting factor and the precoding matrixs of transceivers are jointly optimized. The authors not only analyze the convergence of the proposed algorithm, but also the computational complexity. Some minor issues need to addressed by the authors.

(1) Some newest articles on energy efficient networks are missed in the references. The authors can refer to some latest work such as

"Energy-Efficient Wireless Powered Secure Transmission with Cooperative Jamming for Public Transportation" published in IEEE transaction on green communications and networking in 2019.

"Design of incentive scheme using contract theory in energy-harvesting enabled sensor networks" published in physical communications in 2018.

(2) The title of the subsection 3.1 should be corrected.

(3) As to the channel coefficient matrix given below the 335th line on page 13, it seems ignoring the power attenuation when the signal propogates through the channel. Can the authors explain why?

Round 2

Reviewer 1 Report

OK. I have now checked the response from the authors.
So, my decision is: ACCEPT.